# New Insights on Saporin Resistance to Chemical Derivatization with Heterobifunctional Reagents

**DOI:** 10.3390/biomedicines11041214

**Published:** 2023-04-19

**Authors:** Massimo Bortolotti, Francesco Biscotti, Andrea Zanello, Andrea Bolognesi, Letizia Polito

**Affiliations:** Department of Medical and Surgical Sciences-DIMEC, General Pathology Section, Alma Mater Studiorum University of Bologna, 40126 Bologna, Italy

**Keywords:** cancer therapy, chemical derivatization, conjugates, heterobifunctional reagents, immunotoxins, plant toxins, ribosome-inactivating proteins, rRNA N-glycosylases, saporin, targeted toxins

## Abstract

Saporin is a type 1 ribosome-inactivating protein widely used as toxic payload in the construction of targeted toxins, chimeric molecules formed by a toxic portion linked to a carrier moiety. Among the most used carriers, there are large molecules (mainly antibodies) and small molecules (such as neurotransmitters, growth factors and peptides). Some saporin-containing targeted toxins have been used for the experimental treatment of several diseases, giving very promising results. In this context, one of the reasons for the successful use of saporin lies in its resistance to proteolytic enzymes and to conjugation procedures. In this paper, we evaluated the influence of derivatization on saporin using three heterobifunctional reagents, namely 2-iminothiolane (2-IT), N-succinimidyl 3-(2-pyridyldithio)propionate (SPDP) and 4-succinimidyloxycarbonyl-α-methyl-α-[2-pyridyldithio]toluene (SMPT). In order to obtain the highest number of inserted -SH groups with the lowest reduction of saporin biological activities, we assessed the residual ability of saporin to inhibit protein synthesis, to depurinate DNA and to induce cytotoxicity after derivatization. Our results demonstrate that saporin maintains an excellent resistance to derivatization processes, especially with SPDP, and permit us to define reaction conditions, in which saporin biological properties may not be altered. Therefore, these findings provide useful information for the construction of saporin-based targeted toxins, especially with small carriers.

## 1. Introduction

Ribosome-inactivating proteins (RIPs) are toxic enzymes widespread in the plant kingdom. Regarding their structure, RIPs are classified as type 1, consisting of a single-chain protein with enzymatic activity, and type 2, composed of an enzymatic A-chain linked through a disulfide bond to a lectin B-chain. The presence of the B-chain usually makes type 2 RIPs highly cytotoxic [1,2,3].

RIP enzymatic activity is defined as rRNA N-glycosylase (EC 3.2.2.22), and thanks to this activity, RIPs can remove a specific adenine from rRNA, thus irreversibly damaging ribosomes and causing protein synthesis inhibition [4,5]. Furthermore, RIPs show N-glycosylase activity on other substrates besides rRNA, such as mRNA, tRNA, DNA, poly(A) and poly(ADP-ribose) chains. For this reason, RIP activity has been better defined as polynucleotide: adenosine glycosylase (PNAG) [6,7]. Acting on different substrates, RIPs can trigger multiple cell death pathways and kill cells even if their mechanisms of proliferation and apoptosis are altered, which is common in cancer cells [8,9].

When a toxin is linked to carriers, such as hormones, growth factors, neurotransmitters, small peptides and other molecules, it is commonly referred to as targeted toxin. In particular, a toxin linked to an intact antibody, or a fragment thereof, generates a new molecule, commonly named immunotoxin.

Saporin, the best-known and -characterized type 1 RIP, has been extensively studied as a targeted toxin and as a component of immunotoxins, mainly for cancer therapy [10,11].

The carrier and the toxin can be linked through a chemical bond or fused by genetic engineering to obtain recombinant conjugates. Although the recombinant immunotoxins have demonstrated promising results in vitro and in vivo, poor stability in solution and low protein yield were observed [12].

So far, several saporin-containing immunotoxins have been developed through chemical derivatization and some of them have entered clinical trials, showing highly promising results in the treatment of hematological tumors [13,14,15]. In addition to antibodies, saporin has been also conjugated to small peptides for the experimental therapy of epilepsy, sleep disorders and cancer pain [16,17]. Saporin has been chosen as toxic payload because of its resistance to chemical derivatization and conjugation processes. Moreover, saporin is able to maintain its enzymatic activity at high temperatures, after denaturation with urea and guanidine and after digestion with proteolytic enzymes [18].

The properties of the chemical linker between carrier and toxin should be carefully considered in the targeted toxin design. The linker should not impair the target-binding affinity of the carrier, should be stable in the plasma and should be able to release the toxic payload into the target cells. In particular, disulfide bridge is the most commonly used chemical bond because it exists in nature between the A- and B-chain of plant toxins and because it is efficiently reduced by disulfide isomerase into the cell [19]. This type of chemical bond is obtained upon reaction of the carrier and toxin, previously derivatized using heterobifunctional reagents, in order to introduce reactive thiol groups (-SH) [20]. Among the most frequently used heterobifunctional reagents, there are 2-iminothiolane (2-IT), N-succinimidyl 3-(2-pyridyldithio)propionate (SPDP) and 4-succinimidyloxycarbonyl-α-methyl-α-[2-pyridyldithio]toluene (SMPT).

The reagent 2-IT has been successfully used for derivatization of several RIPs and carriers [21], because it reacts in the 7–10 pH range with the primary amines of the side chains of basic amino acids, forming an amidine bond [22] and inserting a molecular spacer of 8.1 Å [23]. SPDP introduces a spacer of 6.8 Å, reacting with amino groups to form amide bonds, while at the other end of the molecule a disulfide bond is protected by a pyridine-2-thione group [24,25,26]. Likewise, SMPT reacts with the side amino groups of the protein, inserting a thiol group protected by a pyridine-2-thione molecule at the end of a long spacer of 11.2 Å [24,26]. The length and high flexibility of the linker introduced with SMTP make it potentially useful for RIP conjugation to small carriers, avoiding the steric hindrance of the binding site consequent to chemical coupling.

Despite the wide use of these reagents, little is known about the reactivity of saporin to them, the linearity of the chemical derivatization, the number of inserted -SH groups and the resistance of saporin enzymatic and cytotoxic activities to derivatization.

In this paper, we investigated the possibility of obtaining the highest number of inserted -SH groups, which entails as a consequence the lowest reduction of saporin biological activities. For this purpose, we evaluated the effect on saporin of 2-IT, SPDP and SMTP by assessing the ability of derivatized saporin to inhibit protein synthesis, to remove adenines from DNA and to induce in vitro cytotoxicity in comparison with native saporin. This work provides important information to exploit the conjugation potential of saporin, especially when the aim is to chemically link saporin to small carrier molecules.

## 2. Materials and Methods

### 2.1. Materials

Saporin, also known as saporin-S6, was purified from the seeds of *Saponaria officinalis* L., as previously described [27,28]. All the reagents used for derivatization and cell cultures were purchased by Sigma Aldrich (St. Louis, MO, USA), unless otherwise specified. The Sephadex G-25 Coarse gel-filtration resin was provided by Cytiva (Marlborough, MA, USA).

### 2.2. Cells

The NB100 cell line, derived from a human primary neuroblastoma, was from our departmental cell collection [29] and was originally provided by the Laboratory of Pediatric Oncology of the University of Bologna. Cells were cultured at 37 °C in humidified atmosphere at 5% CO_2_ in RPMI 1640 supplemented with 10% fetal calf serum, 2 mM L-Glutamine, 100 units/mL penicillin and 0.1 mg/mL streptomycin, hereinafter defined “complete medium”. To subculture or to seed cells for experiments, the medium was removed and the cell monolayer was washed with 0.14 M NaCl, 5 mM NaH_2_PO_4_ pH 7, Ca^2+^/Mg^2+^ free (PBS). After 5 min of incubation with Trypsin/EDTA (200 mg/L EDTA, 500 mg/L Trypsin), cells were harvested and centrifuged at 500× *g* for 5 min at room temperature. The pellet was resuspended in complete medium and the required number of cells was seeded in flasks or plates.

### 2.3. Derivatization of Saporin with 2-IT

The reagent 2-IT was dissolved immediately prior to use in 50 mM sodium borate buffer pH 9, to final concentrations of 1.75, 2.0 and 2.5 mM. Then, 20 μL of 2-IT solution was added to 2.5 mg of saporin, dissolved in 50 mM sodium borate buffer pH 9, at a final concentration of 7.91 mg/mL. The reaction mixture was incubated at 28 °C in agitation for 90 min. In this way, 2-IT reacts with basic aminoacidic residues (mainly lysine) on saporin surface, thus inserting mercaptobutyrimidoyl linker with free -SH groups (Figure 1). Afterwards, in order to block the reaction, solid glycine was added to a final concentration of 200 mM. After a further 15 min-incubation, -SH groups inserted with 2-IT were protected using 5,5′-dithiobis-(2-nitrobenzoic acid) (Ellman’s reagent), dissolved in *N*,*N*-dimethylformamide (DMF) to a final concentration of 2.5 mM. Ellman’s reagent was added in a volume of 2 μL, mixing the protein solution on a vortex to avoid protein precipitation. After incubation at 28 °C for 5 min, the reaction mixture was applied to a Sephadex G-25 Coarse column, equilibrated and eluted with PBS, to separate the derivatized saporin from unreacted reagents.

The first eluted peak, corresponding to derivatized saporin, was collected following the absorbance at 280 nm. The number of inserted -SH groups was determined by the differential absorbance at 412 nm, before and after reduction with 22 mM β-mercaptoethanol (βME) and the consequent release of 2-nitro-5-thiobenzoic acid (TNB) (Figure 1). The molar concentration of inserted -SH groups was calculated as reported in step 1, Table 1, considering that the molar extinction coefficient at 412 nm of TNB (ε_412(TNB)_) is 14,150 L cm^−1^ M^−1^ [30].

The contribution of TNB at 280 nm was calculated as reported in step 2, Table 1, multiplying the value obtained in step 1 by 2100 L cm^−1^ M^−1^ (molar extinction coefficient at 280 nm of TNB, ε_280(TNB)_) [30].

The protein concentration was calculated as reported in step 3, Table 1, subtracting the above calculated contribution of TNB at 280 nm from the total absorbance at 280 nm, and dividing the saporin contribution by the molar extinction coefficient at 280 nm of saporin (ε_280(saporin)_) that is 24,000 L cm^−1^ M^−1^ [2]. The derivatization ratio, corresponding to the number of -SH groups introduced per protein molecule, was calculated dividing the value obtained from step 1 by that obtained from step 3 (step 4, Table 1).

### 2.4. Derivatization of Saporin with SPDP and SMPT

SPDP and SMPT were dissolved immediately prior to use in DMF to a final concentration of 0.71 mM. The other concentrations (0.57 mM and 0.48 mM) were obtained by diluting the stock solution. Then, 2 μL of such solutions were added to 1.5 mg of saporin, dissolved in 50 mM sodium borate buffer pH 9, at a final concentration of 7.91 mg/mL. The reaction mixture was incubated at 28 °C in agitation for 45 min (SPDP) or for 90 min (SMPT). Due to the higher reactivity of SPDP compared to 2-IT and SMPT, a shorter reaction time was chosen in the case of SPDP.

The choice of different equivalents for 2-IT and SPDP was based on our long-lasting experience. SMPT equivalents were the same as SPDP because these two linkers share the same N-hydroxysuccinimide ester-based chemical reactivity.

In this way, the reagents react with basic aminoacidic residues on saporin surface, thus inserting linkers with a protected -SH group (Figure 2). Afterwards, in order to block the reaction, solid glycine was added to a final concentration of 200 mM. After incubation at 28 °C for 5 min, the reaction mixture was applied to a Sephadex G-25 Coarse column, equilibrated and eluted with PBS, to separate the derivatized saporin from unreacted reagents.

The first eluted peak, corresponding to derivatized saporin, was collected following the absorbance at 280 nm. The number of inserted -SH groups was determined by the differential absorbance at 343 nm, prior and after reduction with 22 mM βME (Figure 2). The reduction with βME results in the release of pyridine-2-thione, whose concentration was determined by measuring the absorbance at 343 nm.

The molar concentration of inserted -SH groups was calculated as reported in step 1, Table 2, considering that the molar extinction coefficient at 343 nm of pyridine-2-thione, (ε_343(pyridine-2-thione)_), is 8080 L cm^−1^ M^−1^ [31].

The contribution of pyridine-2-thione at 280 nm was calculated as reported in step 2, Table 2, multiplying the value obtained in step 1 by 5100 L cm^−1^ M^−1^ (molar extinction coefficient at 280 nm of pyridine-2-thione ε_280(pyridine-2-thione)_) [31].

The protein concentration was calculated as reported in step 3, Table 2, subtracting the above calculated contribution of pyridine-2-thione at 280 nm from the total absorbance at 280 nm, and dividing the saporin contribution by the molar extinction coefficient at 280 nm of saporin (ε_280(saporin)_) that is 24,000 L cm^−1^ M^−1^ [2]. The derivatization ratio, corresponding to the number of -SH groups introduced per protein molecule, was calculated dividing the value obtained from step 1 by that obtained from step 3 (step 4, Table 2).

### 2.5. Cell-Free Protein Synthesis Inhibition Assay

The inhibitory activity on protein synthesis of native saporin and saporin derivatized with 2-IT, SPDP or SMPT was determined using a cell-free rabbit reticulocyte lysate system, as described in [32]. Briefly, the total reaction volume was 62.5 μL, containing 25 μL of RIP, 25 μL of lysate, 10 mM Tris-HCl buffer pH 7.4, 100 mM ammonium acetate, 2 mM magnesium acetate, 1 mM ATP, 0.2 mM GTP, 15 mM creatine phosphate, 12 μU creatine phosphokinase, 0.05 mM amino acids (minus leucine) and 0.75 μCi L-[4,5-^3^H] leucine. The reaction mixture was incubated at 28 °C. After 8 min, first 1 mL of 1 M KOH and then 2 drops of H_2_O_2_ were added. After incubation at 28 °C for 10 min, the total amount of protein was precipitated by adding 1 mL of 20% trichloroacetic acid (TCA) and kept at 4 °C for 60 min. The precipitate was collected on Whatman GF/C discs (Cytiva), using a filter apparatus and washed three times with 1 mL of cold 5% TCA. Radioactivity was counted in 10 mL of Ultima Gold scintillation fluid (Perkin Elmer, Waltham, MA, USA) and determined by a β-counter (Beckman Coulter, Brea, CA, USA). Each experiment was carried out in duplicate and the concentration of RIP causing 50% inhibition of protein synthesis in respect of untreated controls (IC_50_) was calculated by linear regression analysis.

### 2.6. Polynucleotide: Adenosine Glycosylase (PNAG) Activity on Herring Sperm DNA Assay

Herring sperm DNA (10 μg) was incubated with 3 μg of native saporin and saporin derivatized with 2-IT, SPDP or SMPT in 300 μL of a reaction mixture containing 1 M KCl, 0.5 M sodium acetate pH 4.5, at 30 °C for 60 min. After incubation, DNA was precipitated with 100% ethanol and 3 M sodium acetate pH 5.2 at −80 °C overnight, as reported in [33]. The solution was centrifugated at 10,000× *g* at 4 °C for 30 min. Adenines released in the supernatant from RIP-treated DNA were determined spectrophotometrically at 260 nm. Each experiment was carried out in duplicate.

### 2.7. Cell Viability

Cell viability was evaluated using a colorimetric assay (CellTiter 96^®^ Aqueous One Solution Cell Proliferation) (Promega, Woods Hollow Road, Madison, WI, USA).

NB100 cells (3 × 10^3^/well) were seeded in 96-well microtiter plates in 100 μL of complete medium at 37 °C. After 24 h, cells were incubated with native saporin and saporin derivatized with 2-IT, SPDP or SMPT at scalar concentrations in complete medium. After 72 h of incubation at 37 °C, 20 μL/well of kit solution were added. After 1 h of incubation, the absorbance at 492 nm was measured by a microtiter plate reader Multiskan EX (Thermo Labsystems, Helsinki, Finland) [34]. Each experiment was carried out in quadruplicate and the effective concentration of RIP reducing 50% of viability (EC_50_) in NB100 cells was calculated by linear regression, analyzing the dose-response experiments in the linear range of the curves.

### 2.8. Statistical Analysis

Statistical analyses were conducted using XLSTAT-Pro software, version 6.1.9 2003 (Addison, Inc., Brooklyn, NY, USA), using a 95% confidence interval. Results were presented as means ± SD of three different experiments. Data were analyzed using ANOVA test or Mann–Whitney U test.

## 3. Results

### 3.1. Derivatization of Saporin

To obtain the highest number of chemically inserted -SH groups while preserving the enzymatic saporin activity, we performed a detailed study comparing the reactivity of 2-IT, SPDP and SMTP.

In our experiments, the derivatization procedures with 2-IT allowed the insertion of 1.71, 1.98 and 2.52 -SH groups per molecule of saporin, using a ratio (reagent/saporin, mol/mol) of 8, 9 and 11, respectively (Table 3), after a reaction time of 90 min. An excellent linearity (R^2^ = 1) between the concentration of 2-IT and the inserted number of -SH groups was observed.

When saporin was derivatized with SPDP, the inserted numbers of -SH groups were 1.91, 2.23 and 3.02, using a ratio (reagent/saporin, mol/mol) of 2, 2.5 and 3, respectively (Table 3), after a reaction time of 45 min. Also in this case, a good linearity was obtained (R^2^ = 0.987).

The derivatization with SMPT allowed the insertion of 1.20, 1.45 and 1.60 -SH groups per molecule of saporin, using a ratio (reagent/saporin, mol/mol) of 2, 2.5 and 3, respectively (Table 3), after a reaction time of 90 min. After derivatization with SMPT, a good linear trend was observed (R^2^ = 0.929).

Protein derivatization usually generates a mixture of products, on the basis of both the number of available basic aminoacidic residues and the reactivity of the linker. The three reagents showed different reactivities toward saporin. In particular, the levels of derivatization using 2-IT and SPDP were approximately double than those obtained with SMPT.

Moreover, to better define the different products of derivatization reactions, we calculated the percentage of the various molecular classes of derivatized saporin through Poisson distribution (Appendix A).

### 3.2. Effect of Derivatized Saporin on Cell-Free Protein Synthesis Inhibition

The inhibitory activity of saporin derivatized with 2-IT, SPDP or SMPT on cell-free protein synthesis was assayed in vitro, using a rabbit reticulocyte lysate system, and it was compared to native saporin activity (Figure 1). Native saporin strongly inhibited protein synthesis, with IC_50_ of 22 pM.

After derivatization with 2-IT, saporin showed a loss of inhibitory activity directly proportional to the number of inserted -SH groups. Indeed, saporin bearing 1.71 and 1.98 -SH groups showed IC_50_ values of 103 and 143 pM, respectively. Saporin bearing 2.52 -SH groups showed a higher loss of protein synthesis inhibition activity, with IC_50_ value > 173 pM, which corresponds to the highest concentration tested (Table 4).

Saporin derivatization with SPDP did not significantly alter protein synthesis inhibitory activity at any of the tested conditions, with curves and IC_50_ values very close to that of native saporin (Figure 1 and Table 4). Interestingly, SPDP inserted a high number of thiol groups in the saporin molecule without altering its rRNA N-glycosylase activity.

Saporin derivatized with SMPT retained a good ability to inhibit protein synthesis, with IC_50_ values of the same order of magnitude as native saporin (pM) (Table 4), with slight loss of activity and different slope of inhibitory curves (Figure 1).

These results demonstrate that derivatization with 2-IT highly affected rRNA N-glycosylase activity of saporin, with a loss of activity ranging from 4.68 to > 7.86 folds with respect to native saporin. SPDP did not affect saporin inhibitory activity on cell-free protein synthesis, and the loss of inhibitory activity was very small, ranging from a 1.14 to 1.27-fold decrease. Derivatization with SMTP altered the inhibition capacity of protein synthesis in a very limited way, showing a loss of activity ranging from a 1.18 to 1.59-fold decrease, as compared to native saporin (Appendix A).

### 3.3. Effect of Derivatized Saporin on Polynucleotide:Adenosine Glycosylase (PNAG) Activity

PNAG activity of saporin, native and after derivatization with 2-IT, SPDP or SMPT, was spectrophotometrically determined by measuring the release of adenines from herring sperm DNA. As shown in Figure 2, after derivatization with the reagents, a significant reduction of PNAG activity was observed with respect to native saporin, the higher insertion of -SH groups determining the stronger reduction of PNAG activity. Interestingly, using the reagent SPDP, the insertion of 1.91 -SH groups did not significantly alter PNAG activity, compared to native saporin. It should be emphasized that, unlike the loss of rRNA N-glycosylase activity, PNAG activity was reduced to a much lesser extent, with a loss of activity ranging from 1.03 to 1.40 folds for all the reagents and conditions (Appendix A).

### 3.4. Effect of Derivatized Saporin on NB100 Cell Viability

The NB100 cell line was chosen for its high sensitivity to type 1 RIPs, thus allowing the detection of minimal variation in cytotoxicity [29].

The cytotoxicity of saporin derivatized with 2-IT, SPDP or SMPT was assayed on NB100 cells, through dose-response experiments after 72 h of incubation, and was compared to native saporin cytotoxicity (Figure 3). As expected, native saporin displayed the highest cytotoxic effect, with EC_50_ of 259 pM (Table 4).

Derivatization of saporin with 2-IT significantly altered saporin cytotoxic activity (Figure 3). In fact, although the EC_50_ values were of the same order of magnitude as that of native saporin (Table 4), there was a loss of activity of 4.94 (1.71 -SH groups), 6.25 (1.98 -SH groups) and 10.9 (2.52 -SH groups) folds, when compared to native saporin (Appendix A).

Unlike 2-IT, SPDP and SMPT altered the saporin cytotoxic activity to a lesser extent, showing a loss of cytotoxicity with respect to native saporin ranging from 1.45 to 2.86 folds for SPDP and from 1.97 to 2.99 folds for SMPT.

Altogether these results suggested that, at all the tested conditions and for all the reagents, derivatized saporin maintained EC_50_ values in the pM range.

## 4. Discussion

To date, about forty saporin-containing immunotoxins have been synthesized and tested in the experimental therapy of several diseases, as saporin is one of the most used type 1 RIPs in the construction of targeted toxins [14,15,35,36]. The reason for the use of saporin as toxic payload mainly lies in its physico-chemical and biological properties. First of all, the accessibility of the active site to the substrates leads to a high enzymatic activity [37,38]. Second, the resistance to plasmatic and endosomal proteases contributes to saporin stability in vivo, both in blood and inside the cell [18,39]. Furthermore, the available data regarding saporin-containing immunotoxins obtained through chemical derivatization provide us information about the good maintenance of saporin biological properties, after the conjugation procedures. However, it still lacks a systematic and comparative analysis of saporin resistance to the chemical derivatization processes. To improve the knowledge about these aspects, we evaluated the maintenance of saporin enzymatic and cytotoxic activity after derivatization with three of the most commonly used heterobifunctional reagents, namely 2-IT, SPDP and SMPT.

According to the literature, the concentration of 2-IT, probably the most used cross-linking reagent for saporin conjugation, has been usually chosen with the aim to insert about one -SH group per saporin molecule [40,41]. This choice was made to preserve protein activity as much as possible and to have the majority of conjugation product close to one saporin molecule per carrier molecule [9,32,42,43,44]. This approach, based on 1:1 RIP: carrier conjugates, is considered by many researchers to be more suitable than those based on large molecular complexes, which poorly penetrate tumors, mainly solid ones. However, the insertion of more than one -SH group per RIP molecule can be advantageous when the RIP is chosen as payload for small carriers, i.e., neurotransmitters, growth factors and small peptides. In this case, even if saporin is less sterically hindered by the carrier, it becomes necessary to insert a higher number of carrier molecules to increase the binding capacity to the target cell.

Protein derivatization usually generates a mixture of products, on the basis of the number of available basic aminoacidic residues, the reactivity of the linker and the experimental conditions. This is of considerable importance because a heterogeneous mixture of conjugates is generated by the reaction. As type 1 RIPs have many basic available residues, it is not possible to obtain homogeneous reaction products as a consequence of the derivatization process. Furthermore, various RIPs react in different ways to heterobifunctional reagents. The aim of the present work was to clarify, at least for saporin, the reactivity to some of the most common reagents, in terms of linearity of the reaction and possibility of introducing more groups without, or with minimal, inactivation of the molecule. The better understanding of these aspects will allow scientists to control the derivatization and conjugation reactions, in order to minimize the produced conjugate species, to avoid the formation of high molecular weight aggregates and to obtain highly reproducible conjugation conditions.

Our data demonstrate that the derivatization of saporin with the three chosen reagents shows a very good linearity in the tested concentration range, making the reaction very easy to control. SPDP and 2-IT are the most reactive reagents, allowing the insertion of up to 3.02 and 2.52 -SH groups, respectively.

In our experiments, native saporin showed a good protein synthesis inhibitory activity in a cell-free system with an IC_50_ in accordance with values reported in literature [14,15,45]. At all the tested conditions, saporin derivatized with 2-IT showed IC_50_ values significantly higher than those obtained with native saporin. Using 2-IT, the charge of lysyl groups of the protein is maintained; this seems to be an important factor in derivatizing some RIPs, such as gelonin and momordin. In fact, in our experience, such RIPs precipitated and lost activity after derivatization with SPDP, which decreases the surface charge. On the contrary, our data demonstrate an opposite sensitivity of saporin to the selected linkers. It is likely that the charge maintenance when saporin is derivatized with 2-IT is not fundamental, probably due to the high surface positive charge density of saporin [46].

SPDP-derivatized saporin showed IC_50_ values very close to those of native saporin. These results are in agreement with previously reported data, which demonstrated that SPDP-derivatized saporin maintained good activity, whereas other type 1 RIPs showed variable loss of activity [23].

SMPT showed a lower reactivity with respect to SPDP, despite the fact that both reagents possess the same N-hydroxysuccinimide reactive group. This difference in reactivity is quite surprising. A possible reason may lie in the different spacer chemistry, both in terms of length and chemical composition. In particular, the presence of a benzene ring and a longer spacer in SMPT, in comparison to SPDP, could not allow a correct interaction with saporin, whose surface is highly charged.

In addition to the rRNA N-glycosylase activity, saporin, along with many other type 1 and type 2 RIPs, causes the removal of adenines from various biochemical targets, including herring sperm DNA, viral nucleic acids, tRNA and poly(A) polynucleotides [7]. Saporin derivatized with 2-IT, SPDP or SMPT displayed a strong resistance to derivatization processes, maintaining a high PNAG activity in all the tested conditions. In particular, saporin derivatized with the lowest concentration of SPDP (0.48 mM) did not show any significative difference with native saporin, despite the insertion of 1.91 -SH groups.

Besides the evaluation of enzymatic activity in cell-free systems, the cytotoxicity of derivatized saporin was also tested in NB100 cell line. This cell line is ideal to study RIP cytotoxicity, because it is particularly sensitive to both type 1 and type 2 RIPs, thus allowing the detection of minimal variation in cytotoxicity [29]. In our experimental model, SPDP and SMPT caused a slight loss of saporin cytotoxicity (<3 folds); on the contrary, 2-IT caused a higher loss of cytotoxicity (between about 5 to 11 folds, with respect to native saporin).

In conclusion, our data demonstrate that saporin has an excellent resistance to derivatization processes, using heterobifunctional reagents with different chemistry and reactivity. In addition, we demonstrate that saporin is able to retain its enzymatic properties, even with the insertion of more than one -SH group per RIP molecule, a key feature to produce targeted toxins with more than one molecule of small size. In this regard, SPDP proved to be the best heterobifunctional reagent, as it allowed the insertion of the highest number of -SH groups, without substantially altering saporin enzymatic and cytotoxic activities. Even if in our preliminary study we did not carry out tests of conjugation to carrier molecules, the obtained results allow us to predict with a high degree of approximation that the selected linkers inserted the reactive groups on accessible sites of saporin surface. In fact, after the derivatization procedures, saporin maintained its cytotoxic and enzymatic activities, suggesting that no or limited alterations in the active site are present. Taken together, our results give valuable information for the construction of saporin-based targeted toxins, mainly when it is necessary to push the derivatization/conjugation procedures beyond the usual.

## Data Availability

The data supporting the findings of this paper are available on request from the corresponding author.

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
