# Peer review of "New Insights on Saporin Resistance to Chemical Derivatization with Heterobifunctional Reagents"

_biomedicines, 2023, doi:10.3390/biomedicines11041214_

Round 1
Reviewer 1 Report
This paper presents a useful piece of work describing the functional efficacy of the RIP saporin following conjugation procedures involving three different conjugation reagents, nameley SPDP, SMPT and 2IT. The quality of the work is very good with clear descriptions of the methods used together with results obtained from each experiment. I find the paper hard to fault.
The authors have however in my opinion ommtted some key references in support of their paper. The primary paper for the 2IT reagent from King et al (1978) Biochem 17(8), 1499-1506. The paper from Thorpe et al describing the first use of SPDP and SMPT to construct an IT (Thorpe et al (1987) Cancer Res 47, 5924-5931. The paper by Flavell et al comparing the in vitro and in vivo efficacy of anti-CD7 saporin ITs constructed with SMPT and SPDP (Flavell et al Int J Cancer 58, 407-414.
Author Response
This paper presents a useful piece of work describing the functional efficacy of the RIP saporin following conjugation procedures involving three different conjugation reagents, nameley SPDP, SMPT and 2IT. The quality of the work is very good with clear descriptions of the methods used together with results obtained from each experiment. I find the paper hard to fault.
Answer: We thank the reviewer for their appreciation to our paper.
The authors have however in my opinion ommtted some key references in support of their paper. The primary paper for the 2IT reagent from King et al (1978) Biochem 17(8), 1499-1506. The paper from Thorpe et al describing the first use of SPDP and SMPT to construct an IT (Thorpe et al (1987) Cancer Res 47, 5924-5931. The paper by Flavell et al comparing the in vitro and in vivo efficacy of anti-CD7 saporin ITs constructed with SMPT and SPDP (Flavell et al Int J Cancer 58, 407-414.
Answer: We thank the reviewer for their suggestion. All the suggested references have been added to the Introduction (see new references 21, 24 and 25).
Reviewer 2 Report
The authors provide a systematic study how to derivatize the ribosome-inactivating protein saporin without loss of function. This is of high importance because targeted toxins become increasingly important for advanced treatment with biologicals. The authors demonstrated that saporin maintains an excellent resistance to the derivatization processes, especially with N-succinimidyl 3-(2-pyridyldithio)propionate (SPDP), and permit to define reaction conditions, in which saporin biological properties may not be altered. The manuscript is clearly written and understandable. The design of the experiments is conclusive and suitable controls have been conducted. There are, nevertheless, some aspects that should be considered before final publication to make this good manuscript even better.
Specific comments:
1. Line 44–53: There is regrettably no official nomenclature for targeted toxins, so that terms such as “chimeric toxins”, “conjugates”, and “fusion proteins” are not used uniformly in the literature. Therefore, it cannot be defined what is correct and what is not in many cases. Nevertheless, in biochemistry, “conjugates” arise from the linkage of two (preformed) molecules. Thus, in this sense, the term “conjugate” describes the formation of a molecule and not the kind of its components. So, we have either recombinant proteins that are genetically fused (which are produced in one synthesis step) or we have conjugates formed by linkage of two molecules. Thus, immunotoxins can be both, either recombinant fusion proteins or conjugates. Consequently, the term conjugate should preferably not be used to discriminate between immunotoxins and other targeted toxins because both can be produced in one step or by bioconjugation. Thus, the expression “targeted toxins” is the generic term for immunotoxins, growth factor toxins, cytokine toxins, hormone toxins, …
2. Section 2.7 and 3.4. The common determination of EC50 values is done by a four-parameter fit of a sigmoidal curve. Indeed, the sigmoidal curve is quasi-linear close to the inflection point. The sigmoidal course of the curve can be well seen in Fig. 3 (the slope is greater between 1E–10 and 1E–09). Due to missing values at 1E–12 and 1E–07, the sigmoidal course is not optimally visible, however, a four-parameter fit regularly provides more reliable results than linear regression. Nevertheless, for the presented results, a linear fit appears to be sufficient, but the authors should justify in the manuscript, why they used linear regression instead of four-parameter fit.
3. Section 2.8. The authors should explain why they used Student’s t test here. This test requires normal distribution, but how can they assume a normal distribution for the low number of values? Why didn’t they use the Mann-Whitney U test? The authors should explain why they used Student’s t test and should also provide information whether they used the paired or unpaired test.
4. Paragraph lines 225–234: This is introduction and should be moved to it.
5. Lines 235–247: The authors should explain their rationale why they used different equivalents (8, 9, 11 versus 2, 2.5, 3) for the different linkers and why the used equivalents are in such a narrow range. Without any other information (not provided in the manuscript) equivalents of for instance 2, 5, 10 or 20 would be expected from the view of a chemist.
6. Table 5 is superfluous, as it contains only equivalent information to Table 4. Table 5 can be removed without loss of any information.
7. Results: The structure of the results section is confusing because the information in Table 4 is provided before it is explained later in section 3.3 and 3.4. I recommend restructuring the results section in that Table 4 is moved to the end and used to explain the big picture at the end of the results section.
8. Results / discussion: Regrettably, the authors did neither experimentally test one example ligand in their study nor discuss their accessibility for -SH groups and potential effect on functionality of saporin. It can happen that the modification with the least effect on function (SPDP) is the one with worst accessibility for a ligand. Optimally, the authors experimentally show accessibility for their different saporin derivates, e.g., in the easiest way by adding a dye with maleimide function. At least, the authors should explicitly discuss this issue.
9. Discussion: The manuscript is dealing with a very important issue, namely the conjugation of protein toxins to ligands and possible loss of function. This matter has been very well studied. But there is a second point of high importance for chemical conjugates, heterogeneity, which can affect physico-chemical properties such as solubility and also pharmacokinetik behavior. Therefore, newest developments deal with site-specific modifications by click chemistry e.g., by introducing artificial amino acids or sugar side chains with azide functionality. The authors did not discuss the heterogeneity of their products. They only provide average values but did neither give information on the range on modified amino acids, e.g., zero to four, nor on how many different amino acids are modified. The authors should discuss the aspect of heterogeneity in detail and also refer to site-specific reactions within this content.
Author Response
Comments and Suggestions for Authors
The authors provide a systematic study how to derivatize the ribosome-inactivating protein saporin without loss of function. This is of high importance because targeted toxins become increasingly important for advanced treatment with biologicals. The authors demonstrated that saporin maintains an excellent resistance to the derivatization processes, especially with N-succinimidyl 3-(2-pyridyldithio)propionate (SPDP), and permit to define reaction conditions, in which saporin biological properties may not be altered. The manuscript is clearly written and understandable. The design of the experiments is conclusive and suitable controls have been conducted. There are, nevertheless, some aspects that should be considered before final publication to make this good manuscript even better.
Answer: We thank the reviewer for their appreciation to our paper and for their valuable comments and suggestions that helped us to improve our manuscript. Accordingly, the manuscript has been significantly rewritten and it is now better focused and balanced.
Specific comments:
- Line 44–53: There is regrettably no official nomenclature for targeted toxins, so that terms such as “chimeric toxins”, “conjugates”, and “fusion proteins” are not used uniformly in the literature. Therefore, it cannot be defined what is correct and what is not in many cases. Nevertheless, in biochemistry, “conjugates” arise from the linkage of two (preformed) molecules. Thus, in this sense, the term “conjugate” describes the formation of a molecule and not the kind of its components. So, we have either recombinant proteins that are genetically fused (which are produced in one synthesis step) or we have conjugates formed by linkage of two molecules. Thus, immunotoxins can be both, either recombinant fusion proteins or conjugates. Consequently, the term conjugate should preferably not be used to discriminate between immunotoxins and other targeted toxins because both can be produced in one step or by bioconjugation. Thus, the expression “targeted toxins” is the generic term for immunotoxins, growth factor toxins, cytokine toxins, hormone toxins, …
Answer: As suggested, we tried to clarify the nomenclature for targeted toxins and immunotoxins, rewriting lines 45-51 of Introduction, as follow: “When a toxin is linked to carriers, such as hormones, growth factors, neurotransmitters, small peptides and other molecules, it is commonly referred to as targeted toxin. In particular, a toxin linked to an intact antibody, or a fragment thereof, generates a new molecule, commonly named immunotoxin. Saporin, the best-known and -characterized type 1 RIP, has been extensively studied as targeted toxin and as a component of immunotoxins, mainly for cancer therapy”. Moreover, the use of “targeted toxins” term has been used throughout the text where appropriate.
- Section 2.7 and 3.4. The common determination of EC50 values is done by a four-parameter fit of a sigmoidal curve. Indeed, the sigmoidal curve is quasi-linear close to the inflection point. The sigmoidal course of the curve can be well seen in Fig. 3 (the slope is greater between 1E–10 and 1E–09). Due to missing values at 1E–12 and 1E–07, the sigmoidal course is not optimally visible, however, a four-parameter fit regularly provides more reliable results than linear regression. Nevertheless, for the presented results, a linear fit appears to be sufficient, but the authors should justify in the manuscript, why they used linear regression instead of four-parameter fit.
Answer: As suggested, we have justified the use of the linear regression analysis in section 2.7 and in section 3.4 (legend to Figure 3), adding the following sentences: “Each experiment was carried out in quadruplicate and the effective concentration of RIP (EC50) reducing 50% of viability in NB100 cells was calculated by linear regression, analyzing the dose-response experiments in the linear range of the curves.” (section 2.7) and “This statistical approach was used to analyze the dose-response experiments in the linear range of the curves.” (section 3.4, legend to Figure 3).
- Section 2.8. The authors should explain why they used Student’s t test here. This test requires normal distribution, but how can they assume a normal distribution for the low number of values? Why didn’t they use the Mann-Whitney U test? The authors should explain why they used Student’s t test and should also provide information whether they used the paired or unpaired test.
Answer: We thank the reviewer for their valuable suggestion. The Mann-Whitney U test was used instead of Student’s t test. Significance has been recalculated and reported in the new Figure 2. Moreover, the new statistical analysis has been specified in Materials and Methods section (see Paragraph 2.8).
- Paragraph lines 225–234: This is introduction and should be moved to it.
Answer: As suggested, the paragraph was moved to introduction (lines 79-89).
- Lines 235–247: The authors should explain their rationale why they used different equivalents (8, 9, 11 versus 2, 2.5, 3) for the different linkers and why the used equivalents are in such a narrow range. Without any other information (not provided in the manuscript) equivalents of for instance 2, 5, 10 or 20 would be expected from the view of a chemist.
Answer: The choice of different equivalents for two of the linkers used in this paper (2-IT and SPDP) was based on our long-lasting experience (about 140 different conjugates). We used the same equivalents for SPDP and SMPT because they share the same chemical reactivity to proteins. But, SMPT showed a much less reactivity with respect to SPDP already inducing a loss of RIP activity, thus we decided that testing a higher number of equivalents was not necessary.
To better explain this concept, we added new sentences in Materials and Methods (lines 166-168) “The choice of different equivalents for 2-IT and SPDP was based on our long-lasting experience. SMPT equivalents were the same of SPDP because these two linkers share the same N-hydroxysuccinimide ester-based chemical reactivity.” and in Discussion (lines 448-453): “SMPT showed a lower reactivity with respect to SPDP, despite both reagents possess the same N-hydroxysuccinimide reactive group. This difference in reactivity is quite surprising. A possible reason may lie on the different spacer chemistry, both in terms of length and chemical composition. In particular, the presence of a benzene ring and a longer spacer in SMPT, in comparison to SPDP, could not allow a correct interaction with saporin, whose surface is highly charged.”
- Table 5 is superfluous, as it contains only equivalent information to Table 4. Table 5 can be removed without loss of any information.
Answer: As suggested, the table 5 was removed from the manuscript: However, we retain that information in table 5 can be useful for readers and we decided to add it as Supplementary Materials (Table S2).
- Results: The structure of the results section is confusing because the information in Table 4 is provided before it is explained later in section 3.3 and 3.4. I recommend restructuring the results section in that Table 4 is moved to the end and used to explain the big picture at the end of the results section.
Answer: As suggested, we moved Table 4 at the end of the Results section.
- Results / discussion: Regrettably, the authors did neither experimentally test one example ligand in their study nor discuss their accessibility for -SH groups and potential effect on functionality of saporin. It can happen that the modification with the least effect on function (SPDP) is the one with worst accessibility for a ligand. Optimally, the authors experimentally show accessibility for their different saporin derivates, e.g., in the easiest way by adding a dye with maleimide function. At least, the authors should explicitly discuss this issue.
Answer: We thank the reviewer for the valuable suggestion. We consider the advice of adding a dye with a maleimide function very smart and useful; we will certainly perform it in the next steps of this experimental work. However, saporin has 24 lysines, mainly located on the outer surface of the molecule (PDB: 1QI7), that well react with SPDP. Since these lysines are superficial and accessible, it is highly probable that the -SH groups inserted on them will be available too.
As suggested by the reviewer, we added new sentences at the end of the Discussion (lines 475-480): “Even if in our preliminary study we did not carry out tests of conjugation to carrier molecules, the obtained results allow us to predict with a high degree of approximation that the selected linkers inserted the reactive groups on accessible sites of saporin surface. In fact, after the derivatization procedures, saporin maintained its cytotoxic and enzymatic activities, suggesting that no or limited alterations in the active site are present.”
- Discussion: The manuscript is dealing with a very important issue, namely the conjugation of protein toxins to ligands and possible loss of function. This matter has been very well studied. But there is a second point of high importance for chemical conjugates, heterogeneity, which can affect physico-chemical properties such as solubility and also pharmacokinetik behavior. Therefore, newest developments deal with site-specific modifications by click chemistry e.g., by introducing artificial amino acids or sugar side chains with azide functionality. The authors did not discuss the heterogeneity of their products. They only provide average values but did neither give information on the range on modified amino acids, e.g., zero to four, nor on how many different amino acids are modified. The authors should discuss the aspect of heterogeneity in detail and also refer to site-specific reactions within this content.
Answer: We thank the reviewer very much for their comments and valuable suggestions.
To discuss the aspect of heterogeneity, the following paragraph was added in the Discussion (lines 414-426): “Protein derivatization usually generates a mixture of products, on the basis of the number of available basic aminoacidic residues, the reactivity of the linker and the experimental conditions. This is of considerable importance because a heterogeneous mixture of conjugates is generated by the reaction. As type 1 RIPs have many basic available residues, it is not possible to obtain homogeneous reaction products as a consequence of derivatization process. Furthermore, various RIPs react in different manner to heterobifunctional reagents. The aim of present work was to clarify, at least for saporin, the reactivity to some of the most common reagents, in terms of linearity of the reaction and possibility of introducing more groups without, or with minimal, inactivation of the molecule. The better understanding of these aspects will allow scientists to control the derivatization and conjugation reactions, in order to minimize the produced conjugate species, to avoid the formation of high molecular weight aggregates and to obtain highly reproducible conjugation conditions.”
We have inserted in Supplementary materials a new table (Table S1) to clarify that the frequency of the various categories of products follows the distribution of Poisson. Moreover, we have added the following sentences in Results (line 270-277): “Protein derivatization usually generates a mixture of products, on the basis of both the number of available basic aminoacidic residues and the reactivity of the linker. The three reagents showed different reactivities toward saporin. In particular, the levels of derivatization using 2-IT and SPDP were approximately double than those obtained with SMPT. Moreover, to better define the different products of derivatization reactions, we calculated the percentage of the various molecular classes of derivatized saporin through Poisson distribution (Supplementary Table S1).”
Reviewer 3 Report
This topic is interesting and the results can be useful for RIP conjugation. The paper is well written, clear and concise. The discovery and implications are of scientific interest and I would support publication in Pharmaceutics, but there are some issues that need to be addressed.
Line 107 “In this way, 2-IT reacts with arginine and lysine residues on saporin surface..”. Please provide bibliographic evidences of the 2-IT reactivity to arginine. Furthermore, the saporin primary amino group is also able to react.
In Introduction and in line 222 authors stated the importance to obtain the highest number of thiol groups inserted. Currently, in immunotoxin preparation, less than 2 groups provide a good yield of conjugation. The authors should comment how the yield of immunotoxin or other targeted saporin conjugates could be increased by higher thiolation degree. Moreover, authors should discuss about the possible presence of by-products (aggregates) due to high derivatization degree.
Table 3. The authors should describe what is the ratio 2-IT/saporin that allows maintenance of native activity (probably already described in previous papers).
Table 3. Could the authors comment the significative difference in reactivity between SPDP and SMPT although both have an NHS reactive ester?
Results and table 3. How many conjugate replicates have been produced for each reaction? The reported data are the mean of almost three replicates ? So, was the mean values of inserted thiols reproducible?
Figure 1. Please improve the clarity of the figure using a different label (square) or filled circle.
Line 318. The authors should comment the selection of NB100 cell line (sensibility, biological or clinical relevance…)
Discussion. In previous papers the authors always used 2-IT to produce immunotoxins, but from the results of present work SPDP seems to be a better cross-linkage. Please comment.
Discussion. Using 2-IT the charge of lysyl groups is maintained, differently after SPDP or SMTP derivatization. Please discuss this point and the role of surface charge of saporin in relationship to the activity, if known.
Author Response
Comments and Suggestions for Authors
This topic is interesting and the results can be useful for RIP conjugation. The paper is well written, clear and concise. The discovery and implications are of scientific interest and I would support publication in Pharmaceutics, but there are some issues that need to be addressed.
Answer: We thank the reviewer for their appreciation to our paper and for their valuable comments and suggestions that helped us to improve our manuscript. Accordingly, the manuscript has been significantly rewritten and it is now better focused and balanced.
Line 107 “In this way, 2-IT reacts with arginine and lysine residues on saporin surface.”. Please provide bibliographic evidences of the 2-IT reactivity to arginine. Furthermore, the saporin primary amino group is also able to react.
Answer: As suggested, we have added new references, also according to another reviewer’s request (see new references 21, 22, 24 and 25), we have modified the text in the Introduction (Line 81) and Materials and Methods (line 124 and line 169). Moreover, Scheme 1 and 2 have been modified accordingly.
In Introduction and in line 222 authors stated the importance to obtain the highest number of thiol groups inserted. Currently, in immunotoxin preparation, less than 2 groups provide a good yield of conjugation. The authors should comment how the yield of immunotoxin or other targeted saporin conjugates could be increased by higher thiolation degree. Moreover, authors should discuss about the possible presence of by-products (aggregates) due to high derivatization degree.
Answer: The aim of this paper is to define the conditions for linking saporin to small carriers (e.g. growth factors, peptides, neurotransmitters, etc). In these cases, it could be useful to link a higher number of small carrier molecules to the RIP because this will increase cell binding, without excessively increasing the molecular weight of the conjugate and without relevant steric hindrance of the RIP, as we reported in Introduction and Discussion sections. As requested by another reviewer, we discussed the aspect of heterogeneity, by adding the following sentences in the Discussion (lines 414-426): “Protein derivatization usually generates a mixture of products, on the basis of the number of available basic aminoacidic residues, the reactivity of the linker and the experimental conditions. This is of considerable importance because a heterogeneous mixture of conjugates is generated by the reaction. As type 1 RIPs have many basic available residues, it is not possible to obtain homogeneous reaction products as a consequence of derivatization process. Furthermore, various RIPs react in different manner to heterobifunctional reagents. The aim of present work was to clarify, at least for saporin, the reactivity to some of the most common reagents, in terms of linearity of the reaction and possibility of introducing more groups without, or with minimal, inactivation of the molecule. The better understanding of these aspects will allow scientists to control the derivatization and conjugation reactions, in order to minimize the produced conjugate species, to avoid the formation of high molecular weight aggregates and to obtain highly reproducible conjugation conditions.”
Table 3. The authors should describe what is the ratio 2-IT/saporin that allows maintenance of native activity (probably already described in previous papers).
Answer: By our previous experiences, the 2-IT/saporin ratio that allows the maintenance of native saporin activity is close to 1. We have added some references of our previous papers, which show that saporin retains its activity after the insertion of about 1 -SH group per RIP molecule (see references 9, 32, 42 and new references 43, 44).
Table 3. Could the authors comment the significative difference in reactivity between SPDP and SMPT although both have an NHS reactive ester?
Answer: The difference found in reactivity between SPDP and SMPT was quite surprising also for us. In fact, both reagents possess the same reactive ester group. However, SPDP and SMPT have different spacers, both in terms of length and chemical composition, which could determine a different interaction with the protein. In particular, the presence of a benzene ring and a longer spacer in SMPT compared to SPDP could not allow a correct interaction with saporin, whose surface is highly charged. The above explanation has been introduced in Discussion (lines 448-453).
Results and table 3. How many conjugate replicates have been produced for each reaction? The reported data are the mean of almost three replicates? So, was the mean values of inserted thiols reproducible?
Answer: We apologize for the lack of information, we added the number of replicates in the footnotes of Table 3.
Figure 1. Please improve the clarity of the figure using a different label (square) or filled circle.
Answer: As suggested, we improved the clarity of Figures 1 and 3, using square label for derivatized saporin curves.
Line 318. The authors should comment the selection of NB100 cell line (sensibility, biological or clinical relevance…)
Answer: A new sentence was added in Paragraph 3.4, lines 350-351: “The NB100 cell line was chosen for its high sensitivity to type 1 RIPs, thus allowing the detection of minimal variation in cytotoxicity [29].”
Discussion. In previous papers the authors always used 2-IT to produce immunotoxins, but from the results of present work SPDP seems to be a better cross-linkage. Please comment.
Answer: The aim of our previous papers was the conjugation of saporin to antibodies or other large-sized carriers; so, to avoid aggregates and reduce steric hindrance, we had chosen conditions giving an average derivatization ratio close to one. Under these conditions, 2-IT represented a reagent able to satisfy our expectations. The need to conjugate saporin to small carriers prompted us to investigate other experimental conditions and alternative reagents to insert more than one thiol groups per protein molecule.
Discussion. Using 2-IT the charge of lysyl groups is maintained, differently after SPDP or SMTP derivatization. Please discuss this point and the role of surface charge of saporin in relationship to the activity, if known.
Answer: We thank the reviewer for their worthy comment.
Derivatization reaction depends on both the reactivity of the linker and the RIP characteristics. For example, gelonin and momordin usually precipitate during derivatization with SPDP and they lose their activity, while they withstand quite well after derivatization with 2-IT. Other RIPs, such as saporin and PAP, resist very well to chemical modification with SPDP.
We can therefore conclude that for some RIPs the lower reactivity of 2-IT and the conservation of the charge are essential for the maintenance of their structure and activity, while for other RIPs these factors seem irrelevant. However, these aspects have been little investigated and would deserve in-depth studies.
This aspect was commented in the revised Discussion (lines 434-441): “Using 2-IT, the charge of lysyl groups of the protein is maintained; this seems to be an important property to derivatize some RIPs, such as gelonin and momordin. In fact, in our experience, such RIPs precipitated and lost activity after derivatization with SPDP that decrease the surface charge. On the contrary, our data demonstrate an opposite sensitivity of saporin to the selected linkers. Probably, the charge maintenance when saporin is derivatized with 2-IT seems not to be fundamental, probably due to the high surface positive charge density of saporin [46].”
Round 2
Reviewer 2 Report
The authors carefully and adequately revised the manuscript and all concerns raised have been fully addressed. The manuscript is highly important and valuable for a broad readership.
Reviewer 3 Report
The manuscript has been significantly improved and now can be published on Biomedicines.